# Single Layers of Attention Suffice to Predict Protein Contacts

## Abstract

The established approach to unsupervised protein contact prediction estimates co-evolving positions using undirected graphical models. This approach trains a Potts model on a Multiple Sequence Alignment, then predicts that the edges with highest weight correspond to contacts in the 3D structure. On the other hand, increasingly large Transformers are being pretrained on protein sequence databases but have demonstrated mixed results for downstream tasks, including contact prediction. This has sparked discussion about the role of scale and attention-based models in unsupervised protein representation learning. We argue that attention is a principled model of protein interactions, grounded in real properties of protein family data. We introduce a simplified attention layer, *factored attention*, and show that it achieves comparable performance to Potts models, while sharing parameters both within and across families. Further, we extract contacts from the attention maps of a pretrained Transformer and show they perform competitively with the other two approaches. This provides evidence that large-scale pretraining can learn meaningful protein features when presented with unlabeled and unaligned data. We contrast factored attention with the Transformer to indicate that the Transformer leverages hierarchical signal in protein family databases not captured by our single-layer models. This raises the exciting possibility for the development of powerful structured models of protein family databases.

## 1 Introduction

Inferring protein structure from sequence is a longstanding problem in computational biochemistry. Potts models, a particular kind of Markov Random Field (MRF), are the predominant unsupervised method for modeling interactions between amino acids. Potts models are trained to maximize pseudolikelihood on alignments of evolutionarily related proteins (Balakrishnan et al., 2011; Ekeberg et al., 2013; Seemayer et al., 2014). Features derived from Potts models were the main drivers of performance at the CASP11 competition (Monastyrskyy et al., 2016) and have become standard in state-of-the-art supervised models (Wang et al., 2017; Yang et al., 2019; Senior et al., 2020).

Inspired by the success of BERT (Devlin et al., 2018), GPT (Brown et al., 2020) and related unsupervised models in NLP, a line of work has emerged that learns features of proteins through self-supervised pretraining (Rives et al., 2020; Elnaggar et al., 2020; Rao et al., 2019; Madani et al., 2020; Nambiar et al., 2020). This new approach trains Transformer (Vaswani et al., 2017) models on large datasets of protein sequences. There is significant debate over the role of pretraining in protein modeling. Pretrained model performance raises questions about the importance of data and model scale (Lu et al., 2020; Elnaggar et al., 2020), the potential for neural features to compete with evolutionary features extracted by established bioinformatic methods (Rao et al., 2019), and the benefits of transfer learning for protein landscape prediction (Shanehsazzadeh et al., 2020).

In this paper, we take the position that attention-based models can build on the strengths of both Potts trained on alignments and Transformers pretrained on databases. We introduce a simplified model, *factored attention*, and show that it is motivated directly by fundamental properties of protein sequences. We demonstrate empirically that a single layer of factored attention suffices to predict protein contacts competitively with state-of-the-art Potts models, while leveraging parameter sharing across positions within a single family and across hundreds of families. This highlights the potential for explicit modeling of biological properties with attention mechanisms.

Further, we systematically demonstrate that contacts extracted from ProtBERT-BFD (Elnaggar et al., 2020) are competitive with those estimated by Potts models across 748 protein families, inspired by recent work from Vig et al. (2020). We find that ProtBERT-BFD outperforms Potts for proteins with alignments smaller than 256 sequences. This indicates that large-scale Transformer pretraining merits continued efforts from the community.

We contrast factored attention with ProtBERT-BFD to identify a large gap between the gains afforded by the assumptions of single-layer models and the performance of ProtBERT-BFD. This suggests the existence of properties linking protein families that allow for effective modeling of thousands of families at once. These properties are inaccessible to Potts models, yet are implicitly learned by Transformers through extensive pretraining. Understanding and leveraging these properties represents an exciting challenge for protein representation learning.

Our contributions are as follows:

1. We analyze the assumptions made by the attention mechanism in the Transformer and show that a single attention layer is a well-founded model of interactions within protein families; attention for proteins can be justified without any analogies to natural language.

2. We show that single-layer models can successfully share parameters across positions within a family or sharing of amino acid features across hundreds of families, demonstrating that factored attention achieves performance nearly identical to Potts with far fewer parameters.

3. We carefully benchmark ProtBERT-BFD against an optimized Potts implementation and show that the pretrained Transformer extracts contacts competitively with Potts.

## 2  BACKGROUND

Proteins are polymers composed of amino acids and are commonly represented as strings. Along with this 1D sequence representation, each protein folds into a 3D physical structure. Physical distance between positions in 3D is often a much better indicator of functional interaction than proximity in sequence. One representation of physical distance is a *contact map* $C$, a symmetric matrix in which entry $C_{ij} = 1$ if the $\beta$ carbons of $i$ and $j$ are within 8Å of one another, and 0 otherwise.

**Multiple Sequence Alignments.** To understand structure and function of a protein sequence, one typically assembles a set of its evolutionary relatives and looks for patterns within the set. A set of related sequences is referred to as a *protein family*, commonly represented by a Multiple Sequence Alignment (MSA). Gaps in aligned sequences correspond to insertions from an alignment algorithm (Johnson et al., 2010; Remmert et al., 2012), ensuring that positions with similar structure and function line up for all members of the family. After aligning, sequence position carries significant evolutionary, structural, and functional information. See Appendix A.1 for more information.

**Coevolutionary Analysis of Protein Families.** The observation that statistical patterns in MSAs can be used to predict couplings has been widely used to infer structure and function from protein families (Korber et al., 1993; Göbel et al., 1994; Lapedes et al., 1999; Lockless & Ranganathan, 1999; Fodor & Aldrich, 2004; Thomas et al., 2008; Weigt et al., 2009; Fatakia et al., 2009). Let $X_i$ be the amino acid at position $i$ sampled from a particular family. High mutual information between $X_i$ and $X_j$ suggests an interaction between positions $i$ and $j$. The main challenge in estimating contacts from mutual information is to disentangle "direct couplings" corresponding to functional interactions from interactions induced by non-functional patterns (Lapedes et al., 1999; Weigt et al., 2009). State-of-the-art estimates interactions from MRF parameters, as described below.

**Supervised Structure Prediction.** Modern structure prediction methods take a supervised approach, taking MSAs as inputs and outputting predicted structural features. Deep Learning has greatly advanced state of the art for supervised contact prediction (Wang et al., 2017; Jones & Kandathil, 2018; Senior et al., 2020; Adhikari, 2020). These methods train deep residual networks that take covariance statistics or coevolutionary parameters as inputs and output contact maps or distance matrices. Extraction of contacts without supervised structural signal has not seen competitive performance from neural networks until the recent introduction of Transformers pretrained on protein databases.

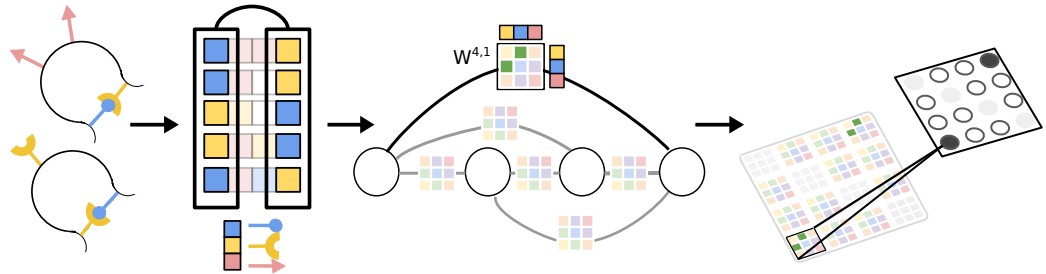

Figure 1: Training and interpretation of a Potts model on a single protein family. The proteins are all loops formed by one blue and one yellow amino acid locking together. The MSA for this family aligns these critical yellow and blue amino acids. For the trained MRF on this MSA, the weight matrix $W^{4,1}$ has the highest values due to the evolutionary constraint that blue and yellow covary for those positions. In this case, the highest predict contact recapitulates a true contact.

**Attention-Based Protein Models.** Elnaggar et al. (2020); Rives et al. (2020); Rao et al. (2019); Madani et al. (2020) train unsupervised attention-based models on millions to billions of protein sequences and show that pretraining can produce good representations. Rao et al. (2019) and Rives et al. (2020) assess embeddings as inputs to supervised structure prediction methods. When protein structure is known, Ingraham et al. (2019) and Du et al. (2019) have used sparse attention based on the physical structure of the protein to directly encode priors about interaction.

## 3 METHODS

Throughout this section, $x = (x_1, \ldots, x_L)$ is a sequence of length $L$ from an alphabet of size $A$. This sequence is part of an MSA of length $L$ with $N$ total sequences. Recall that a fully-connected Pairwise MRF over $p$ variables $X_1, \ldots, X_p$ specifies a distribution

$$p_\theta(x_1, \ldots, x_p) = \frac{1}{Z} \exp \left( \sum_{i<j} E_\theta(x_i, x_j) \right),$$

where $Z$ is the partition function and $E_\theta(x_i, x_j)$ is an arbitrary function of $i$, $j$, $x_i$ and $x_j$. For all models below, we can introduce an explicit functional $E_\theta(x_i)$ to capture the marginal distribution of $X_i$. When introduced, we parametrize the marginal with $E_\theta(x_i) = b_{i,x_i}$ for $b \in \mathbb{R}^{L \times A}$.

### 3.1 POTTS MODELS

A Potts model is a fully-connected pairwise MRF with $L$ variables, each representing a position in the MSA. An edge $(i, j)$ is parametrized with a matrix $W^{ij} \in \mathbb{R}^{A \times A}$. These matrices are organized into an order-4 tensor which form the parameters of a Potts model, see Figure 1. Note that $W^{ij} = W^{ji}$. The energy functional of a Potts model is given through lookups, namely

$$E_\theta(x_i, x_j) = W^{ij}(x_i, x_j). \tag{1}$$

### 3.2 FACTORED ATTENTION

Like Potts, factored attention is a fully-connected pairwise MRF with $L$ variables. The parameters of this model consist of $H$ triples $(W_Q, W_K, W_V)$, where $W_Q, W_K \in \mathbb{R}^{L \times d}$; $W_V \in \mathbb{R}^{A \times A}$; and $d$ is a hyperparameter. Each such triple is called a *head* and $d$ is the *head size*. Unlike a Potts model, the parameters for each edge $(i, j)$ are tied through the use of heads. The energy functional is

$$E_\theta(x_i, x_j) = \sum_{h=1}^{H} \mathrm{symm} \left( \mathrm{softmax} \left( W_Q^h W_K^{h\,T} \right) \right)_{ij} W_V^h(x_i, x_j), \tag{2}$$

where $\mathrm{symm}(M) = (M + M^T)/2$ ensures the positional interactions are symmetric.

Factored attention has two advantages over Potts for modeling protein families: it shares a pool of amino acid feature matrices across all positions and it estimates $\mathcal{O}(L)$ parameters instead of $\mathcal{O}(L^2)$.

**Sharing amino acid features.** Many contacts in a protein are driven by similar interactions between amino acids, such as many types of weakly polar interactions (Burley & Petsko, 1988; Jaenicke, 2000). If two pairs of positions $(i, j)$ and $(l, m)$ are both in contact due to the same interaction, a Potts model must estimate completely separate amino acid features $W^{ij}$ and $W^{lm}$. In order to share amino acid features, we want to compute all energies from one pool of $A \times A$ feature matrices. The simplest way to accomplish this is by associating an $L \times L$ matrix $\mathcal{A}$ to every $A \times A$ feature matrix $W_V$. For $H$ such pairs $(\mathcal{A}, W_V)$, we could introduce a factorized MRF:

$$E_\theta(x_i, x_j) = \sum_{h=1}^{H} \text{symm} \left( \text{softmax} \left( \mathcal{A}^h \right) \right)_{ij} W_V^h(x_i, x_j). \tag{3}$$

A row-wise softmax is taken to encourage sparse interactions and aid in normalization. This model allows the pairs $(i, j)$ and $(l, m)$ to reuse a single feature $W_V^h$, assuming $\mathcal{A}_{ij}^h$ and $\mathcal{A}_{lm}^h$ are both large.

**Scaling linearly in length.** Both Potts and the factorized model in Equation 3 have $\mathcal{O}(L^2)$ parameters. However, contacts are observed to grow linearly over the wide range of protein structures currently available (Taylor & Sadowski, 2011; Kamisetty et al., 2013), which we examine in Figure 11. Given that the number of interactions we wish to estimate grows linearly in length, the quadratic scaling of these models can be greatly improved. One way to fix this is by introducing the factorization $\mathcal{A} = W_Q W_K^T$, where $W_Q, W_K \in \mathbb{R}^{L \times d}$. As before, we employ a row-wise softmax for sparsity and normalization.

Combining feature sharing with linear length scaling leads to Equation 2.

**Recovering attention with sequence-dependent interactions.** All models introduced so far estimate a single undirected graphical model from the training data. While a single graph can be a good approximation for the structure associated with a protein family, many families have *subfamilies* with different functional specializations and even different underlying contacts (Brown et al., 2007; Malinverni & Barducci, 2019). Since subfamily identity is rarely known, allowing edge weights to be a function of sequence could enable the estimation of a family of highly related graphs.

The attention mechanism of the Transformer implements sequence-dependent edge weights by allowing positional interactions to be a function $\mathcal{A}(x)$. In the language of the Transformer, factored attention estimates a single graph because it computes queries and keys using only the positional encoding. More precise explanations are in Section A.2.

While data-dependent interactions significantly increase model capacity, this comes with multiple tradeoffs. Demands on GPU memory increase significantly with data-dependent interactions, as we detail in Section A.3. Further, attention-based models can be very challenging to train (Zhang et al., 2019; Liu et al., 2019); we found both Potts and factored attention trained more stably in all experiments. Lastly, the expressivity conferred by attention creates a risk of overfitting, especially when the pretraining dataset is small. Effective training on small MSAs is particularly important, since MSA depth correlates strongly with contact quality (Schaarschmidt et al., 2018).

**Self-Supervised Losses.** Given an MSA, state-of-the-art methods estimate Potts model parameters through pseudolikelihood maximization (Kamisetty et al., 2013; Ekeberg et al., 2013). On the other hand, BERT-like attention-based models are typically trained with variants of masked language modeling (Devlin et al., 2018). Pseudolikelihood is challenging to compute efficiently for generic models, unlike masked language modeling. Both of these losses require computing conditionals of the form $p_\theta(x_i|x_{\setminus M})$, where $M$ is a subset of $\{1, \ldots, L\} \setminus \{i\}$. The losses $\mathcal{L}_{PL}$ and $\mathcal{L}_{MLM}$ for pseudolikelihood and masked language modeling, respectively, are

$$\mathcal{L}_{PL}(\theta; x) = \sum_{i=1}^{L} \log p_\theta(x_i|x_{\setminus i}), \qquad \mathcal{L}_{MLM}(\theta; x, M) = \sum_{i \in M} \log p_\theta(x_i|x_{\setminus M}).$$

Regularization for Potts and factored attention are both based on MRF edge parameters, while single-layer attention is penalized using weight decay. More details can be found in Section A.4.

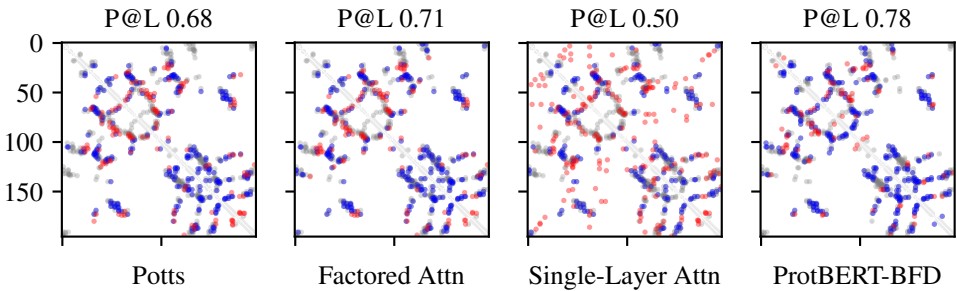

Figure 2: Predicted contact maps and Precision at $L$ for each model on PDB entry *2BFW*. Blue indicates a true positive, red indicates a false positive, and grey indicates a false negative.

### 3.3 EXTRACTING CONTACTS

**Potts.** We follow standard practice and extract a contact map $\widehat{C} \in \mathbb{R}^{L \times L}$ from the order-4 interaction tensor $W$ by setting $\widehat{C}_{ij} = \|W^{ij}\|_F$.

**Factored Attention.** Since factored attention is a pairwise MRF, we can compute its order-4 interaction tensor $W$ and use the same procedure as Potts. See Equation 4.

**Single-Layer Attention.** To produce contacts for an MSA, we compute attention maps from *only* the positional encoding (without sequence) and average attention maps from all heads. Each single-layer attention model is trained on one MSA, so the positional encoding is a feature shared by all sequences in the MSA.

**ProtBERT-BFD.** We extract contacts from ProtBERT by averaging a subset of attention maps for an input sequence $x$. Of the 16 heads in 30 layers, we selected six whose attention maps had the top individual contact precisions over 500 families randomly selected from the Yang et al. (2019) dataset. Predicted contacts for $x$ are given by averaging the $L \times L$ attention maps from these six heads, then symmetrizing additively. See Table 2.

**Average Product Correction (APC).** Empirically, Potts models trained with Frobenius norm regularization have artifacts in the outputs $\widehat{C}$. These are removed with the Average Product Correction (APC) (Dunn et al., 2008). Unless otherwise stated, we apply APC to all extracted contacts.

## 4 RESULTS

**Experimental Setup.** We use a set of 748 protein families from Yang et al. (2019) to evaluate all models. For Potts models and single attention layers, we train separate models on each individual MSA. ProtBERT-BFD is frozen for all experiments. We train models using PyTorchLightning (Falcon, 2019) and Weights and Biases (Biewald, 2020). We compare predicted contact maps $\widehat{C}$ to true contact maps $C$ using standard metrics based on precision. A particularly important metric is *precision at $L$*, where $L$ is the length of the sequence (Schaarschmidt et al., 2018; Shrestha et al., 2019). This is computed by masking $\widehat{C}$ to only consider positions $\geq 6$ apart, predicting the top $L$ entries to be contacts, and computing precision. We provide more information on data and metrics in Appendix A.6 and more information on model hyperparameters in Section A.7.

**Attention assumptions reflected in 15,051 protein structures.** We examine all 15,051 structures in the Yang et al. (2019) dataset for evidence of two key properties useful for single-layer attention models: few contacts per residue and the number of contacts scaling linearly in length. Figure 11 shows a linear trend of number of contacts versus length. In Figure 12, we see that 80% of the 3,747,101 million residues in these structures have 4 or fewer contacts. Only 1.8% of residues have more than ten contacts. This shows that the row-wise softmax, which encourages each residue to attend to only a few other residues per-head, reflects structure found in the data.

**Factored attention matches Potts performance on 748 families.** Figure 2 shows a representative sample of good quality contact maps extracted from all models. Figure 3a summarizes the perfor-

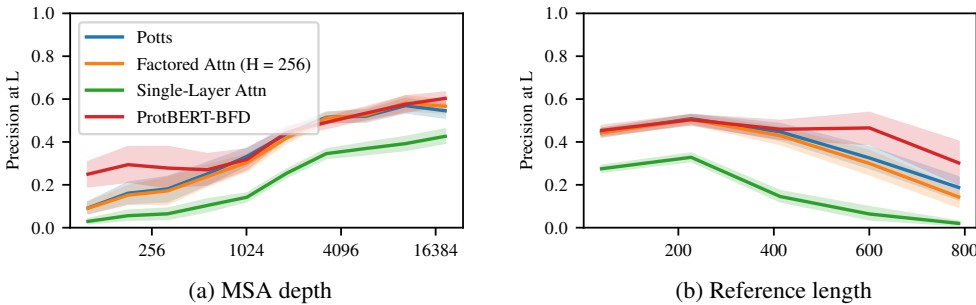

Figure 3: Model performance evaluated on MSA depth and reference length.

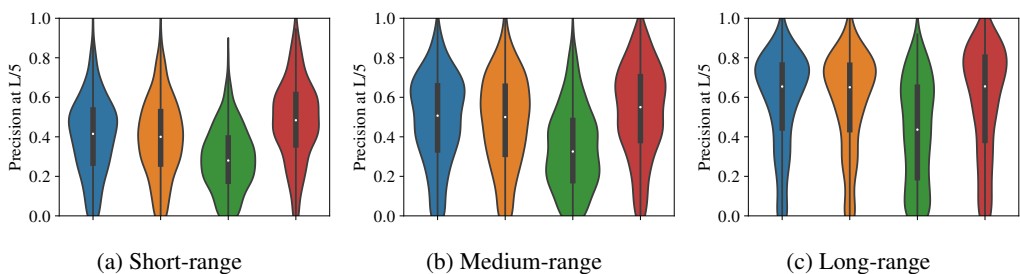

Figure 4: Contact precision for all models stratified by the range of the interaction.

mance of all models over the set of 748 protein families. Factored attention, Potts, and ProtBERT-BFD have comparable overall performance, with median precision at $L$ of 0.46, 0.47, and 0.48, respectively. Stratifying by number of sequences reveals that ProtBERT-BFD has higher precision on MSAs with fewer than 256 sequences. For MSAs with greater than 1024 sequences, Potts, factored attention, and ProtBERT-BFD have comparable performance. Single-layer attention is uniformly worse over all MSA depths.

Next, we evaluate the impact of sequence length on performance. Figure 3b shows that factored attention and Potts achieve similar precision at $L$ over the whole range of family lengths, despite factored attention having far fewer parameters for long families. This shows that factored attention can successfully leverage sparsity assumptions where they are most useful.

Long-range contacts are particularly important for downstream structure-prediction algorithms – long-range precision at $L/5$ is reported in both CASP12 and CASP13 (Schaarschmidt et al., 2018; Shrestha et al., 2019). Figure 4 breaks down contact precisions based on position separation into short ($6 \leq sep < 12$), medium ($12 \leq sep < 24$), and long ($24 \leq sep$). We see that ProtBERT-BFD performs best on short-range contacts, with a median increase of 0.068 precision at $L/5$. On long-range ProtBERT-BFD, there is no appreciable difference in performance to Potts and factored attention. Across the range of contact bins, factored attention and Potts perform very similarly.

**Fewer heads can match Potts on $L/5$ contacts.** We probe the limits of parameter sharing by lowering the number of heads in factored attention and evaluating whether fewer heads can be used to precisely estimate contacts. Figure 5a shows that 128 heads can be used to estimate $L/5$ contacts as precisely as Potts over the full set of 748 families. In Figure 5b, we see that factored attention with 32 and 64 heads is still able to achieve reasonable overall performance compared to Potts. 32 and 64 heads have precision at $L/5$ at least as high as Potts for 329 and 348 families, respectively. If we wish to recover the top $L$ contacts, 256 heads are required to match Potts across all families, as seen in Figure 13. Having more heads than 256 does not further increase performance. Intriguingly, Figure 14 demonstrates that both Spearman and Pearson correlation between the order-4 interaction tensors of factored attention and Potts improve even when increasing to 512 heads. We do not observe the same trends for increasing head size, as shown in Figure 15.

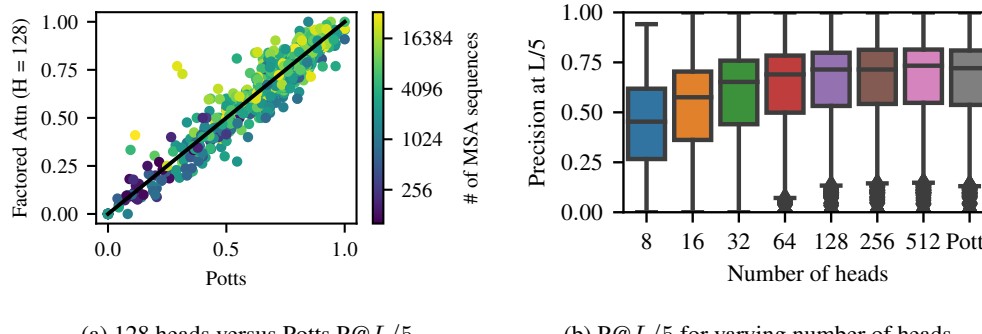

(a) 128 heads versus Potts P@$L/5$

(b) P@$L/5$ for varying number of heads.

Figure 5: Examining impact of number of heads on precision at $L/5$. Left: Comparing performance of Potts and 128 heads over each family shows comparable performance. Right: Precision at $L/5$ drops off slowly until 32 heads, then steeply declines beyond that.

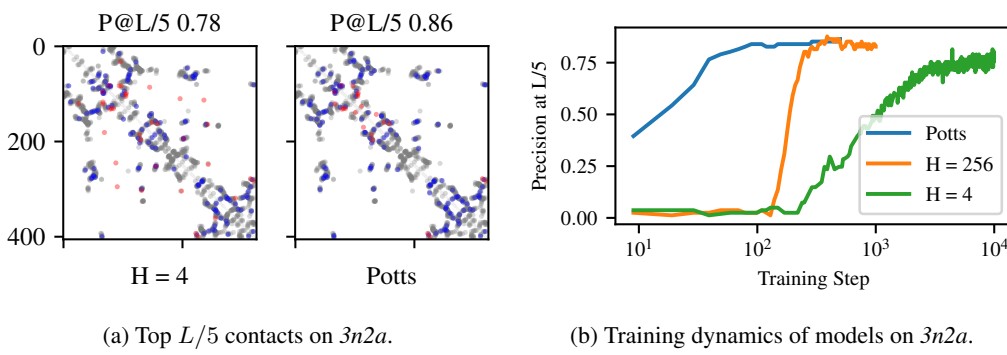

(a) Top $L/5$ contacts on *3n2a*.

(b) Training dynamics of models on *3n2a*.

Figure 6: Factored attention with 4 heads can learn the top $L/5$ contacts on *3n2a*.

For some families, the number of heads can be reduced even further. We show an example on the MSA built for PDB entry *3n2a*. In Figure 6a, we see that merely 4 heads are required to recover $L/5$ contacts nearly identical to those recovered by Potts. This shows that shared amino acid features and interaction parameters can enable identical performance with a 300x reduction in parameters. The training dynamics of these models are shown in Figure 6b. Both factored attention with 256 heads and Potts converge after roughly 100 gradient steps, whereas factored attention with 4 heads requires nearly 10,000 steps to converge. In Figure 16, we show that the top $L$ contacts are significantly worse for 4 heads compared to Potts.

**One set of amino acid features can be used for all families.** Thus far we have only examined models that share parameters within single protein families. Since Prot-BERT is trained on an entire database, it can leverage feature sharing across families to attain greater parameter efficiency and improved performance on small MSAs.

To explore the possibility that attention can share parameters across families, we train factored attention using a single set of frozen value matrices. We first train factored attention normally on *3n2a* with 256 heads, then freeze the learned value matrices for the remaining 747 families. The query and key parameters are trained normally. In Figure 7, we compare the precision at $L$ of factored attention with frozen *3n2a* features to that of factored attention trained normally. Using a single frozen set of features results in only 6 families seeing precision at $L$ decrease by more than 0.05, with a maximum

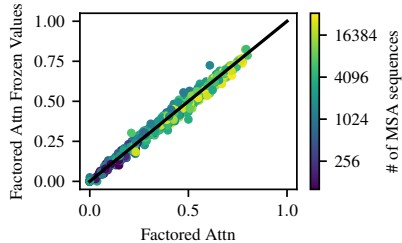

Figure 7: A single set of frozen value matrices can be used for all families.

drop of 0.11. Frozen values are also comparable to Potts performance, as expected – see Figure 17. This suggests that, even for a single-layer model, a single set of value matrices can capture amino acid features across functionally and structurally distinct protein families.

**Factored attention reduces total parameters estimated.** For an MSA of length $L$ with alphabet size $A$, Potts models require $\binom{L}{2}A^2$ parameters. Factored attention with $H$ heads and head size $d$ requires $H(2Ld + A^2)$ parameters. In Figure 18, we plot number of parameters versus length for various values of $H$ and $d = 32$. Potts requires a total of 12 billion parameters to model all 748 families. Factored attention with 256 heads and head size 32 has 3.2 billion parameters; lowering to 128 heads reduces this to 790 million. Half of this reduction comes from 107 families of length greater than 400. ProtBERT-BFD is the most efficient, with 420 million parameters.

**Ablations** APC has a considerable impact on both Potts and factored attention, creating a median increase in precision at $L$ of 0.1 and 0.07, respectively. The effect of APC is negligible for single-layer attention and ProtBERT. Replacing pseudolikelihood maximization with Masked Language Modeling did not appreciably change performance for either Potts or factored attention. Addition of the single-site potential $b^i$ increases performance slightly for attention layers, but not enough to change overall trends. To compare to ProtBERT-BFD, we train our single-layer attention models on unaligned families and found that performance degrades significantly. See Figures 19-22.

## 5 DISCUSSION

We have shown that attention-based models achieve state-of-the-art performance on unsupervised contact extraction and that the assumptions encoded by attention reflect important properties of protein families. These results suggest that attention has a natural role in protein representation learning, without analogy to attention's success in the domain of NLP.

The motivating principles of factored attention were 1) sharing amino acid features across positions and 2) reducing parameters to $\mathcal{O}(L)$. The additional assumption of sequence-dependent contacts led to attention. The parametrizations of these properties presented in this paper were chosen to match those of the Transformer, but many alternatives exist. We believe broader exploration of this space will be fruitful. For example, the importance of APC for factored attention suggests including it directly into the attention mechanism, rather than as a correction after training. The success of Potts and factored attention demonstrates that estimating a single graphical model for a single protein family is often a reasonable approximation. Adapting attention to better model a set of highly related graphs could prove useful for capturing subfamily structure within deep MSAs, or for transferring information across related families. There remains, unexplored, a rich space of protein-specific architectures that potentially have the capacity to learn powerful protein features.

Many mysteries surround the success of Transformers at extracting protein contacts through masked language modeling. Most surprising is the apparent ability of ProtBERT to model many protein families at once, despite not being presented any information about protein family identity or even the existence of such a classification. Beyond family information, ProtBERT appears to learn parsimonious representations which share a vast amount of signal across protein families. Even including a number of within-family hierarchical assumptions *and* sharing amino acid features across all families, factored attention falls far short of ProtBERT in its potential to efficiently represent tens of thousands of protein families. Despite its impressive performance, ProtBERT does not improve long-range contact extraction in our evaluations. Much work remains to understand how pretraining can be most useful for downstream applications.

Our results show that hierarchical structure, both within and across families, is a source of signal available to attention models. Understanding how such structure can be learned without the use of protein family labels could lead to the development of widely applicable modeling components for protein representation learning. We believe that unsupervised attention-based models have the potential to impact structure prediction as broadly as existing coevolutionary methods.

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

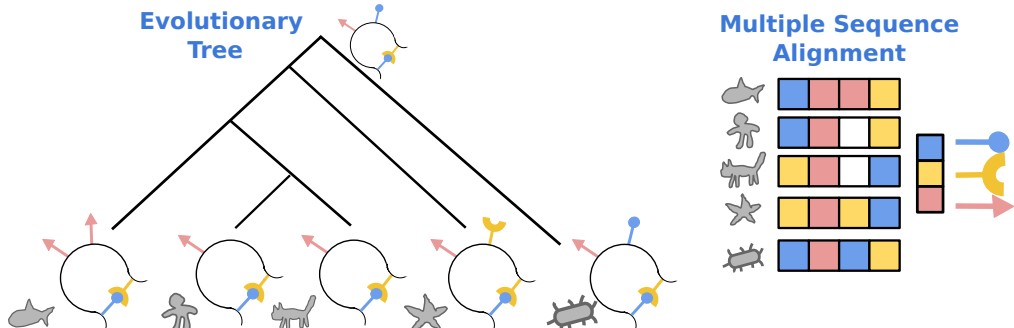

Figure 8: The tree on the right depicts evolution of a protein family. The protein at the root is the ancestral protein, and the five proteins at the leaves are its present-day descendants. The alignment on the left is the corresponding Multiple Sequence Alignment of observed sequences.

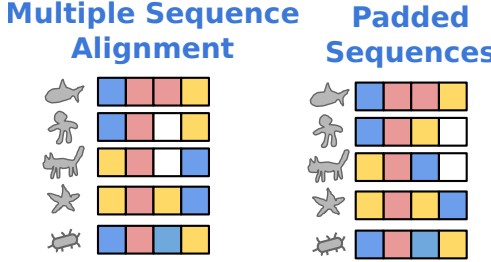

Figure 9: MSA for sequences from Figure 8 compared to a padded batch of the same sequences.

J Vig, A Madani, L R Varshney, C Xiong, and others. Bertology meets biology: Interpreting attention in protein language models. *arXiv preprint arXiv*, 2020.

Sheng Wang, Siqi Sun, Zhen Li, Renyu Zhang, and Jinbo Xu. Accurate de novo prediction of protein contact map by ultra-deep learning model. *PLOS Computational Biology*, 13(1):1–34, 01 2017. doi: 10.1371/journal.pcbi.1005324. URL https://doi.org/10.1371/journal.pcbi.1005324.

M Weigt, R A White, H Szurmant, and others. Identification of direct residue contacts in protein–protein interaction by message passing. *Proceedings of the*, 2009.

Jianyi Yang, Ivan Anishchenko, Hahnbeom Park, Zhenling Peng, Sergey Ovchinnikov, and David Baker. Improved protein structure prediction using predicted inter-residue orientations. *bioRxiv*, pp. 846279, 2019. doi: 10.1101/846279. URL https://www.biorxiv.org/content/10.1101/846279v1.

Jingzhao Zhang, Sai Praneeth Karimireddy, Andreas Veit, Seungyeon Kim, Sashank J Reddi, Sanjiv Kumar, and Suvrit Sra. Why adam beats sgd for attention models. *arXiv preprint arXiv:1912.03194*, 2019.

# A  APPENDIX

## A.1  MULTIPLE SEQUENCE ALIGNMENTS

We demonstrate key concepts of protein evolution in Figure 8. On the left is a *phylogenetic tree*. The leaves represent five observed proteins in a family, while the root represents their most recent common ancestor. The ancestral protein was a loop and evolution preserved this loop structure. Thus every observed sequence has one yellow and one blue amino acid on its ends, which lock together to pinch off the loop. The amino acids in the middle of the loop exhibit considerable differences

within the family, presumably leading to variations in function. This variation within the protein family is captured by the MSA on the right through its placement of gap characters (white squares). Compared to standard padding, shown in Figure 9, the placement of gap characters in the MSA ensures all blue and yellow amino acids lie in a single column. This signal is obfuscated by standard padding.

The problem of multiplying aligning a set of sequences has been extensively studied in Bioinformatics and Computational Biology (Durbin et al., 1998). Given a set of $N$ sequences of average length $\tilde{L}$, a full multiple alignment can be generated by performing dynamic programming on an array of size $\mathcal{O}(\tilde{L}^N)$. This is computationally intractable; the common workaround is to perform iterative pairwise alignment to a fixed reference sequence. The MSAs we use were generated with the HHSuite (Steinegger et al., 2019).

### A.2 Recovering Factored Attention from Standard Attention

We show more precisely that factored attention can be recovered from standard attention one of two ways

1. Computing queries and keys from one-hot positional encodings and values from one-hot sequence embeddings.

2. Using a learned positional embedding and sequence embedding with identity query, key, and value matrices.

The symmetrization operator symm is not applied to Transformer multihead attention, so we present a slight variation of factored attention without symm in this section.

**Single attention layer.** Given a sequence of dense vectors $x = (x_1, \ldots, x_L)$ with $x_i \in \mathbb{R}^p$, the attention mechanism of the Transformer encoder (multihead scaled dot product self-attention) produces a continuous representation $y \in \mathbb{R}^{L \times p}$. If head size is $d$, this representation is computed using $H$ heads $(W_Q, W_K, W_V)$, where $W_Q, W_K, W_V \in \mathbb{R}^{p \times d}$. Queries, keys, and values are defined as $Q = xW_Q, K = xW_K, V = xW_V$. For a single head $(W_Q, W_K, W_V)$, the output is given by

$$y = \text{softmax}\left(\frac{QK^T}{\sqrt{d}}\right)V.$$

The full output in $\mathbb{R}^{dH}$ is produced by concatenating all head outputs. A single Transformer encoder layer passes the output through a dense layer, applying layer-norms and residual connection to aid optimization.

For the first layer, the input $x$ is a sequence of discrete tokens. To produce a dense vector combining sequence and position information, positional encodings and sequence embeddings are combined. The positional encoding $E_{pos} \in \mathbb{R}^{L \times e}$ produces a dense vector of dimension $e$ for each position $i$. The sequence embedding $E_{seq} \in \mathbb{R}^{A \times e}$ maps each element of the vocabulary to a dense vector of dimension $e$. Typically these are combined through summation to produce a dense vector $\tilde{x}_i = E_{seq}(x_i) + E_{pos}(i)$, which is input to the Transformer as described above.

For this paper, we use only multihead self-attention without the dense layer, layernorm, or residual connections, as these drastically hurt performance when employed for one layer.

**Factored attention from standard attention.** Written explicitly, the input Transformer layer computes queries for a single head with $Q = (E_{pos} + E_{seq}(x))W_Q$. Keys and values are computed similarly. To recover factored attention, we instead compute queries and keys via $Q = E_{pos}W_Q$ and $K = E_{pos}W_K$, while values are given by $V = E_{seq}(x)W_V$. For simplicity, we one-hot encode both position and sequence, which corresponds using identity matrices $E_{pos} = I_L \in \mathbb{R}^{L \times L}$ and $E_{seq} = I_A \in \mathbb{R}^{A \times A}$. Equivalently, one can view the positional encoding and sequence embedding as being learned while fixing $W_Q, W_K$, and $W_V$ to be identity matrices.

**Implicit single-site term in single-layer attention.** For a single layer of attention, the product $E_{pos}W_V$ is a matrix in $\mathbb{R}^{L \times A}$. This matrix does not depend on sequence inputs, thus allowing it to act as a single-site term. This suggests why inclusion of an explicit single-site term in Figure 22 had no effect for single-layer attention.

A.3  COMPUTATIONAL EFFICIENCY OF SINGLE LAYER MODELS

| Model | Length | Batches/s |
|---|---|---|
| Potts | 120 | 36.59 |
| | 440 | 5.21 |
| | 904 | 1.04 |
| Factored attention ($H = 256$) | 120 | 32.89 |
| | 440 | 3.91 |
| | 904 | 1.03 |
| Single-layer attention ($H = 128$) | 120 | 17.15 |
| | 440* | 1.685* |
| | 904* | 0.46* |

Table 1: Throughput of various models for batches of size 32. Stars (*) indicate usage of gradient accumulation due to GPU memory constraints. For sequences of length 440, batches of size 16 were used for standard attention. For sequences of length 904, batches of size 4 were used.

In Table 1, we show the number of gradient steps per second for Potts, factored attention, and single-layer attention. We fix a batch size of 32 and report numbers for families of lengths 120, 440, and 904 in order to explore the impact of length on computational efficiency for all models. Metrics are collected on a node with a single NVIDIA RTX 2080Ti, an Intel i7-7800x with 6 physical cores and 12 virtual cores, and 32 GB of system RAM. To get steady state batches per second, we run each model for 5000 steps and report the mean batches per second between steps 500 and 4500. Potts and factored attention have similar throughput, while standard attention scales much worse in length due to its memory requirements necessitating the use of gradient accumulation. This poor memory performance is because standard attention must compute $L \times L$ attention maps for each batch element separately, whereas the $L \times L$ component of other models does not depend on sequence.

A.4  LOSSES

The loss for all three models is of the form

$$\ell(\theta; x) = \mathcal{L}(\theta; x) + cR(\theta),$$

where $\mathcal{L}$ is either pseudolikelihood or masked language modeling and $R$ is a regularizer.

**Potts regularization.** Consider the order-4 interaction tensor $W$, where $W^{ij} \in \mathbb{R}^{A \times A}$ gives the parameters associated to edge $(i, j)$. We regularize $W$ by setting $R(\theta) = \sum_{i<j} \|W^{ij}\|_F^2$. This term is multiplied by $\lambda \cdot L \cdot A$, following Ovchinnikov et al. (2014).

**Factored attention regularization.** Since factored attention is also a fully connected pairwise MRF, we use identical regularization to that of Potts. The order-4 tensor $W$ is given by

$$W_{ab}^{ij} = \sum_{h=1}^{H} \text{symm}\left(\text{softmax}\left(W_Q^h W_K^{h\,T}\right)\right)_{ij} W_V^h(a, b). \tag{4}$$

**Single-layer attention regularization.** Due the lack of an MRF interpretation for single-layer attention, we chose to use weight decay as is typically done for attention models. This corresponds to setting $R(\theta)$ to be the sum of Frobenius norm squared for all weights $W_Q$, $W_K$ and $W_V$.

**Single-site term.** When any model has a single-site term, we follow standard practice and regularize its Frobenius norm squared.

## A.5 PROTBERT-BFD HEAD SELECTION

| layer | head | P@L |
|-------|------|-------|
| 29 | 7 | 0.517 |
| 29 | 8 | 0.396 |
| 29 | 4 | 0.394 |
| 29 | 2 | 0.353 |
| 29 | 11 | 0.333 |
| 29 | 0 | 0.299 |
| 28 | 3 | 0.275 |
| 29 | 15 | 0.177 |
| 29 | 6 | 0.167 |
| 29 | 12 | 0.158 |
| 28 | 4 | 0.141 |
| 29 | 9 | 0.139 |
| 28 | 6 | 0.125 |
| 28 | 5 | 0.125 |
| 3 | 4 | 0.115 |
| 28 | 11 | 0.106 |

Table 2: The top 16 heads in ProtBERT-BFD whose attention maps gave the most precise contact maps across 500 validation families. Most of the top performing heads are found in the last layer. The top six heads were selected for our contact extraction in all results.

## A.6 DATA AND METRICS

### A.6.1 SELECTION OF PROTEIN FAMILIES

We used the following sets of families for model development:

1. A set of 748 families was chosen for performance evaluation. All metrics reported in the paper are on this set, with a single choice of hyperparameters for Potts models, factored attention, and standard attention. The 748 families were chosen randomly from the Yang et al. (2019) dataset, which consists of 15,051 MSAs generated from the databases Uni-Clust30 and UniRef100 (Suzek et al., 2007), as well as metagenomic datasets. Our random sample is representative of the range of MSA depths and protein lengths, see Figure 10.

2. A set of six families from the 748 was chosen to choose hyperparameters for single-layer attention. They were chosen to span a range of MSA depth (large and small), as well as three different regimes of Potts performance (Good, Ok, Poor). These families were used to tune hyperparameters as described in Section A.7.1. See Table 3.

3. A set of ten families from the 748 was chosen where factored attention performed very poorly in our initial experiments. Half were chosen to be long proteins and the other half to be short. This set was used to optimize learning rate and regularization for factored attention to ensure reasonable model performance. See Table 4.

4. 500 entirely disjoint families were further selected randomly from Yang et al. (2019) and used to compute average precision for each head in ProtBERT-BFD (Elnaggar et al., 2020). Performance on these families was used for selecting the top 6 heads, see Table 2.

| PDB | Sequences | Length | Potts Performance |
|-----|-----------|--------|-------------------|
| 3er7_1_A | 33673 | 118 | Good |
| 5fo5_1_B | 17560 | 88 | Ok |
| 2w18_1_A | 33619 | 308 | Poor |
| 4gnr_1_A | 2073 | 351 | Good |
| 5mkc_1_A | 515 | 207 | Ok |
| 3e9l_1_A | 146 | 292 | Poor |

Table 3: 6 families chosen for hyperparameter optimization for single-layer attention.

| PDB | Sequences | Length |
|---|---|---|
| 4k61_1_A | 2145 | 140 |
| 4l3r_1_A | 5535 | 143 |
| 3cy4_1_B | 1064 | 154 |
| 6fdg_1_A | 2325 | 155 |
| 3p6b_1_B | 4353 | 186 |
| 1jm1_1_A | 17130 | 202 |
| 4yt2_1_A | 15481 | 343 |
| 3vmm_1_A | 4383 | 471 |
| 4egc_1_A | 9929 | 539 |
| 3gq7_1_A | 6568 | 605 |

Table 4: 10 families chosen for hyperparameter optimization for factored attention

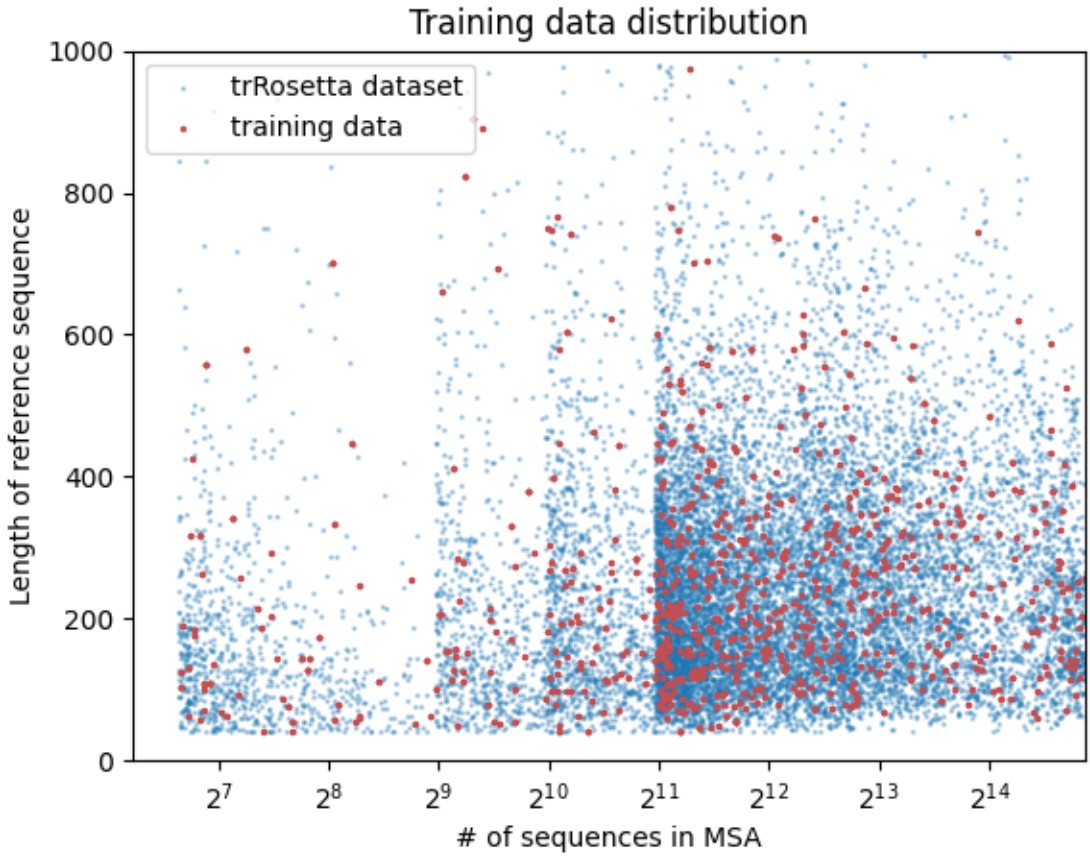

Figure 10: The length and MSA size distribution for our 748 family subset (red) compared to the full 15,051 families in the trRosetta dataset selected for training

### A.6.2    PRODUCING CONTACT MAPS

A PDB structure gives 3D coordinates for every atom in a structure. We use Euclidean distance between the beta carbons to define distance between any pair of positions. A pair of positions where this distance is less than $8\mathring{A}$ is declared to be a contact.

### A.6.3 Scoring Contact Predictions

Given a predicted contact map $\widehat{C} \in \mathbb{R}^{L \times L}$ and a true contact map $C \in \{0, 1\}^{L \times L}$, we describe metrics for scoring $\widehat{C}$.

A sequence $x = (x_1, \ldots, x_L)$ of length $L$ has $\binom{L}{2}$ potential contacts. Since we see $\mathcal{O}(L)$ contacts, contact prediction a sparse prediction task. Accordingly, we focus on precision-recall based quantitative analyses of $\hat{C}$. Common practice in the field is to sort all $\binom{L}{2}$ entries of $\hat{C}$ in decreasing order and evaluate precision at various length thresholds, such as the top $L$ or $L/10$ predictions (Shrestha et al., 2019). Note that this analysis is similar to choosing recall cutoffs along a precision-recall curve, where sorted length index plays the role of recall on the $x$ axis. Unlike recall, length-based cutoffs do not rely on knowledge of the actual number of contacts. In addition to the precision at various length (recall) cutoffs, we also computed Area Under the Precision-Recall Curve (AUC). AUC is a widely used metric for comparing classifiers when the positive class is rare.

### A.7 Hyperparameters

**Potts.** We used $\lambda = 0.5$, learning rate of 0.5, and batch size 4096. Pad, gap, and mask were all encoded with the same token.

The Potts model is trained using a modified version of Adam presented in Dauparas et al. (2019). This modification was made to improve performance of Adam to match that of L-BFGS.

**Factored attention.** We AdamW with a learning rate of 5e-3 and set $\lambda = 0.01$. The default head size was set to 32 unless stated otherwise.

**Single-layer attention.** We set embedding size of 256, head size of 64, and number of heads 128. The model is trained with AdamW using a learning rate of 5e-3 and weight decay of 2e-3. Attention dropout of 0.05 is also applied. The batch size is 32 and mask prob for masked language modeling is 0.15. We use a separate mask token and pad,gap token.

**ProtBERT-BFD.** ProtBERT-BFD has 30 layers each with 16 heads and a hidden size of 1024. The training dataset is a mixture of UniRef50 (Suzek et al., 2015) and BFD. It has $2, 122$ million protein sequences. See Elnaggar et al. (2020) more information.

### A.7.1 Hyperparameter Sweep Details

**Potts.** The Potts model implementation using psuedolikelihood has been optimized by others, so we did not tune performance. Since performance with MLM was comparable to pseudolikelihood, we did not sweep for MLM either.

**Single-layer attention.** Standard attention is by far the most sensitive model to hyperparameters. To find a reasonable set of hyperparameters, we first swept over the six families in Table 3, performing a grid search over

- $H \in \{32, 64, 128, 256, 512\}$
- $d \in \{32, 64, 128, 256, 512\}$
- $e \in \{128, 256, 512\}$
- attention dropout in $\{0, 0.05, 0.1\}$
- learning rate in $\{5e - 3, 1e - 2\}$
- weight decay in $\{0, 1e - 3, 2e - 3\}$

We found that the choice $H = 256$, $d = 64$, $e = 256$, attention dropout of 0.05, learning rate of $5e - 3$ and weight decay of $2e - 3$ performed well across all six families. Due to GPU memory constraints, we had to set $H = 128$ for further runs.

**Factored attention.** We swept factored attention over the families in Table 4, performing a grid search over

- learning rate in $\{1e - 3, 5e - 3, 1e - 2, 5e - 2\}$

- regularization coefficient in $\{1e-4, 5e-4, 1e-3, 5e-3, 1e-2\}$

We found that learning rate of 5e-3 and regularization of 0.01 were effective, but that other configurations such as regularization of 5e-3 also performed well. Both $H$ and $d$ are evaluated extensively in our results.

## A.8   ADDITIONAL FIGURES

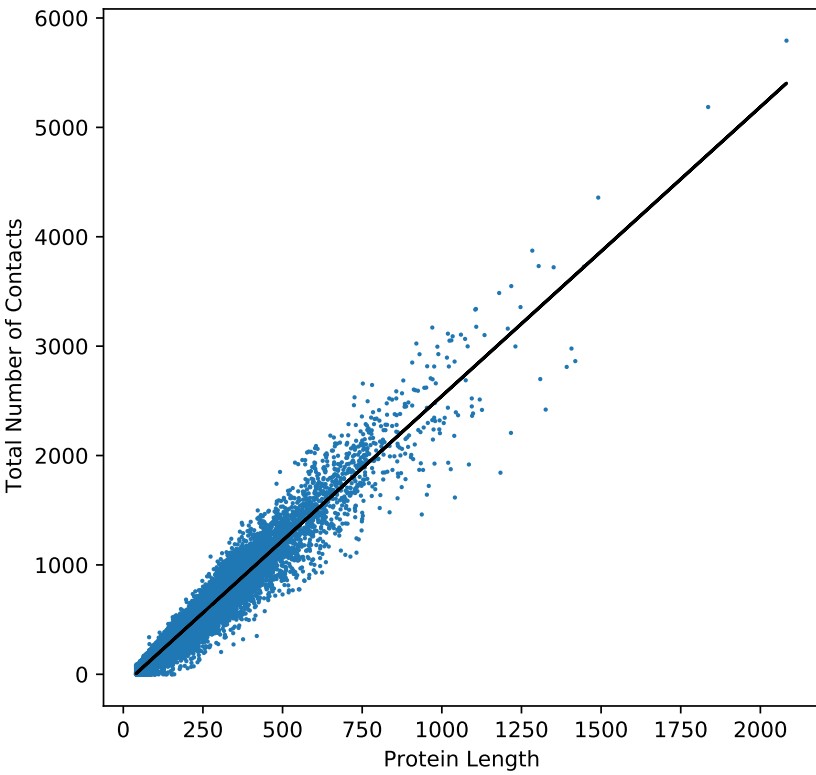

Figure 11: The total number of contacts for a structure as a function of protein length follows a linear trend. (slope $= 2.64$, $R^2 = 0.929$)

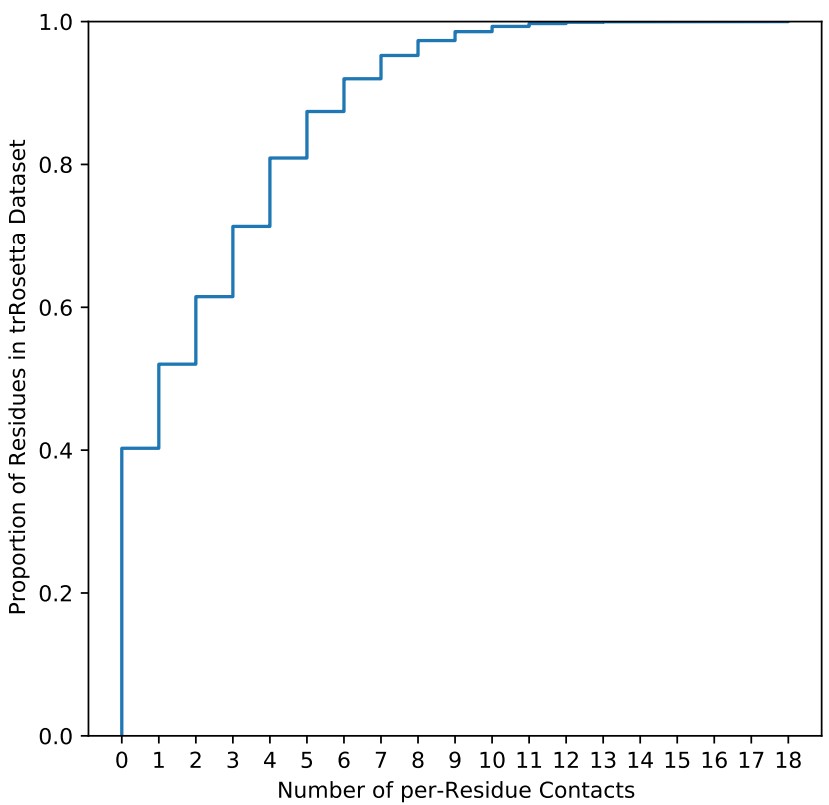

Figure 12: The empirical CDF of number of per-residue contacts for 3,747,101 residues in 15,051 structures in the trRosetta dataset.

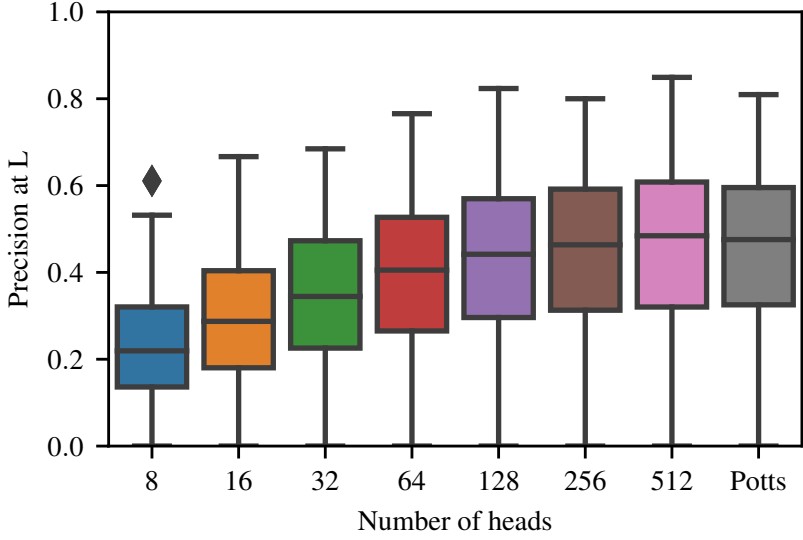

Figure 13: Reducing the number of heads causes a much steeper decrease in precision at $L$.

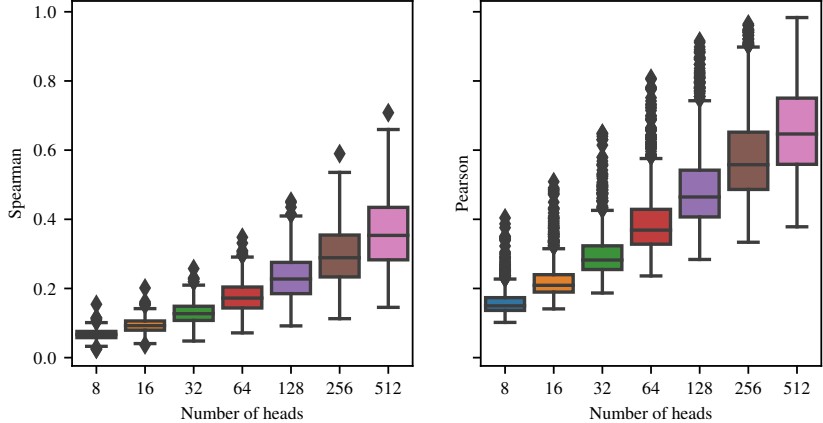

Figure 14: Effect of number of heads on correlation between the order-4 weight tensors for factored attention (see Equation 4) and Potts (see Section 3).

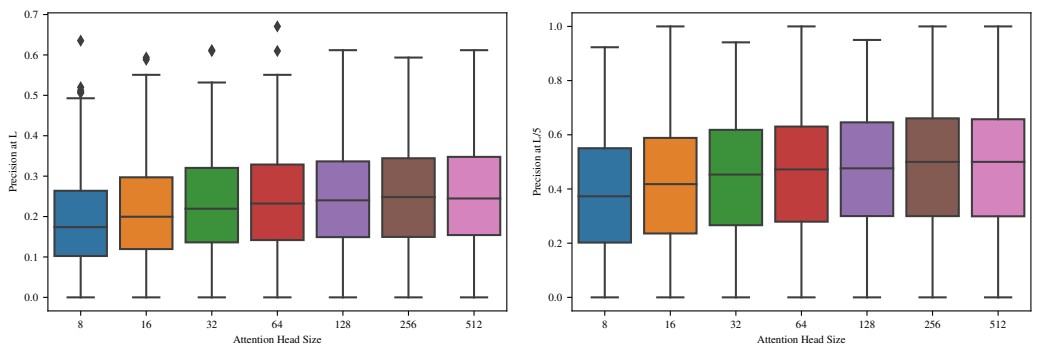

Figure 15: Effect of head size on factored attention precision at $L$ and $L/5$ over 748 families. Increasing head size has a small effect on precision, though not nearly as pronounced as the number of heads.

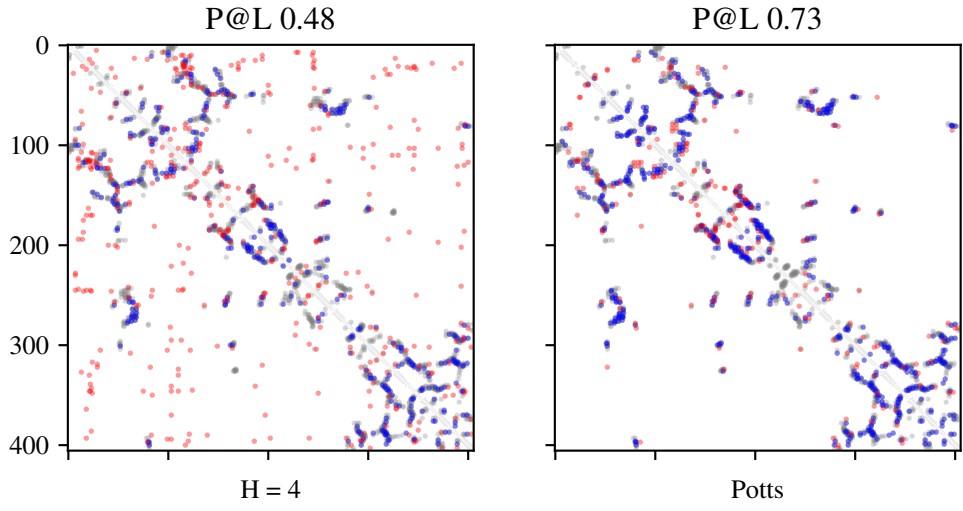

Figure 16: 4 heads has degraded performance for precision at L.

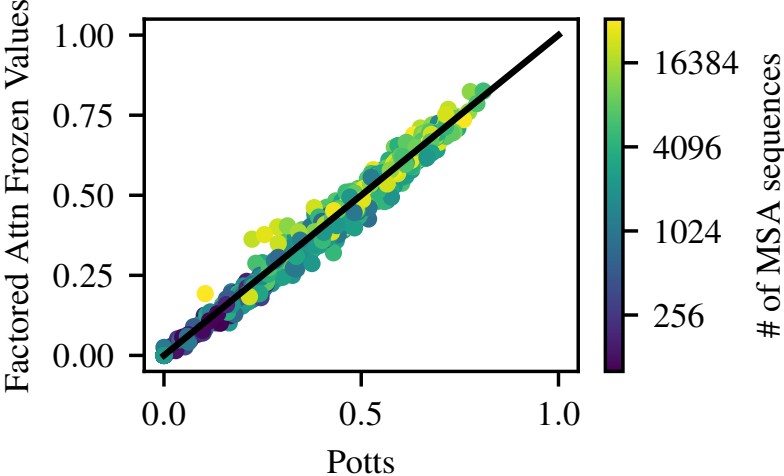

Figure 17: Factored attention trained with a single set of frozen value matrices performs comparably to Potts, evaluated on precision at L across 748 families.

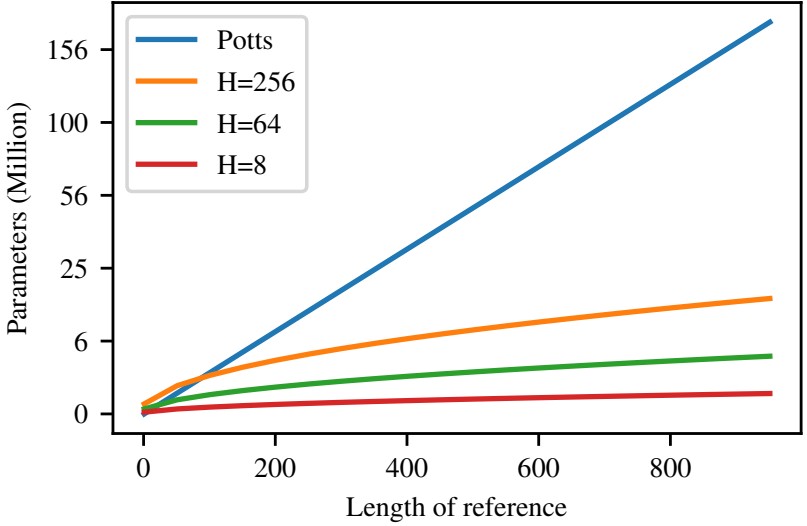

Figure 18: Number of parameters versus length for MRF models.

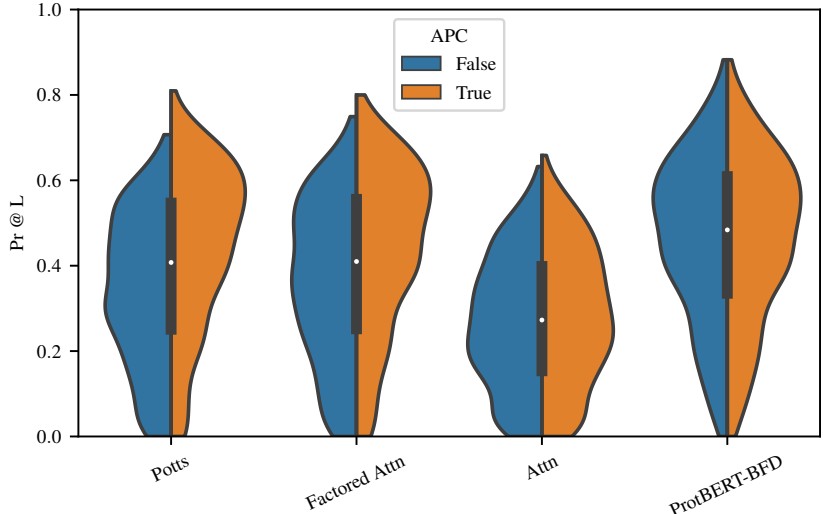

Figure 19: APC has a significant positive effect on the performance of Potts and factored attention. It makes only a slight difference on the performance of the other two models.

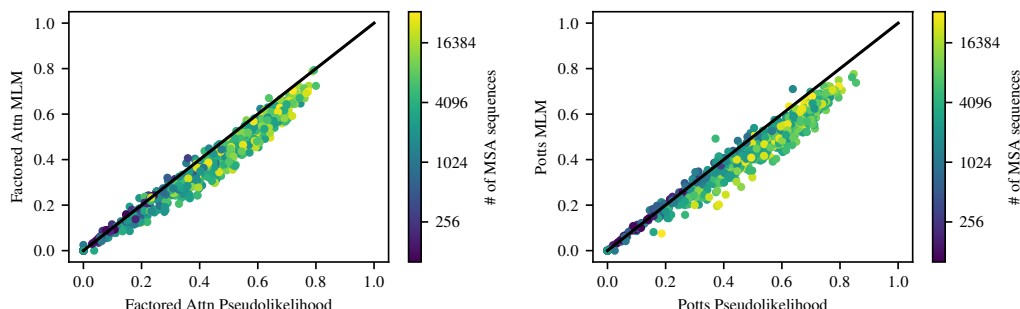

Figure 20: Effect of loss on precision at $L$ over many families. Pseudolikelihood has a uniform but small benefit over masked language modeling for both models.

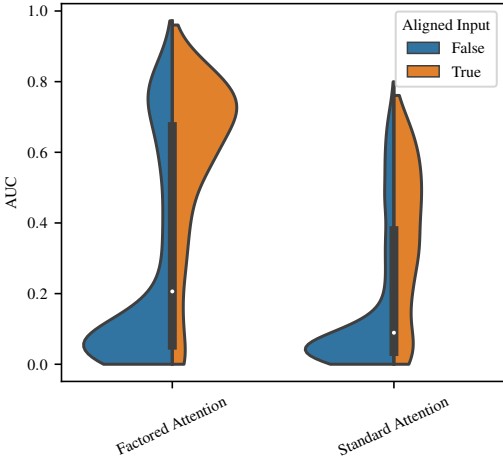

Figure 21: Training on unaligned families degrades performance on almost all families.

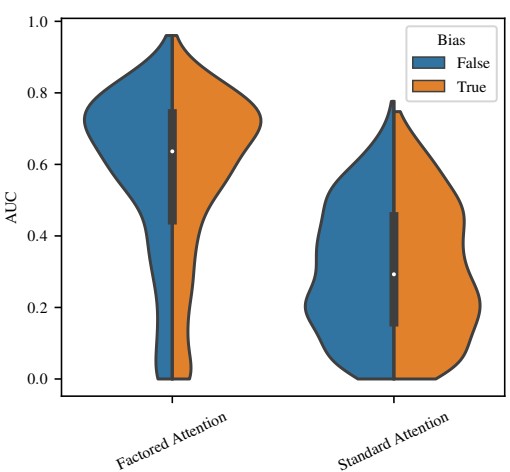

Figure 22: The addition of a single-site term to either factored or standard attention produces little additional benefit.

