# OpenReview forum: "Single Layers of Attention Suffice to Predict Protein Contacts"
_ICLR.cc/2021/Conference — Reject_

### Official Review · AnonReviewer2 · 2020-10-20
**Overall good paper about the relationship between Transformers and Potts models**

**Rating:** 7
**Confidence:** 3

**Review:**

Summary
=======
Transformers models have been recently shown to capture protein contact information in their attention maps when trained unsupervised of millions of protein sequences. This paper draws parallels between Transformers and Potts models (fully-connected pairwise MRF)--the current standard approach for protein contact prediction--and shows empirically that Transformers are competitive with Potts models. Understanding the differences and similarities between Transformers and Potts models makes Transformers less of a ‘black-box’ and helps to establish them as a principled method for contact prediction. The paper is clearly written and the evaluation is solid. I have only a few comments.


Major comments
=============
1. What is the maximum sequence similarity between the training sequence of ProtBERT and sequences in TrRosetts alignments that were used for testing? Sequences must not overlap have a maximum similarity of let’s say 80%.

2. You describe that you used three sets of families from the TrRosetta dataset (A.4.1). Why did you use only 732 families for testing (set 3)? Were these all families that were not included in the first two sets? How many families do the first two sets include and how similar are families of different sets? Ideally, train, tune, and test families belong to different super families.

3. You describe in section A.3 how you extracted protein contact maps from the attention maps of ProtBERT. This is an important detail that must be described in the main text. How did you choose the 6 heads? Did you choose them manually or, for example, by training a linear model to predict contacts from attention maps and using the weights for identifying important heads, or computing the weighted average of attention maps?


Minor comments
=============
4. Section 3.2, ‘x = E_seq(x_i) + E_pos(i)’: How did you compute positional embeddings and why do and add embeddings instead of concatenating them?

5. Section 3.2, ‘We treat the positional embedding E_pos as an overall summary per-position information’. Please describe more clearly what this summary is.

6. Section 4, first paragraph: The L of the precision at L metric is not the sequence length but the number of top sequences. You describe L as being both.

7. Figure 6 is not discussed. Instead of showing this figure, I suggest quantifying the correlation depending on the number of heads by computing and discussing  the Spearman correlation.

8. Rives et al  2020 ‘Biological structure and function emerge…’ have recently shown in addition to Vig et al that protein contact can be predicted from attention maps, which must be also pointed out in the ‘Background’ section.

---

> ### Author Response · Authors · 2020-11-24
> **Response to Reviewer 2**
>
> > The paper is clearly written and the evaluation is solid. I have only a few comments.
>
> We appreciate reviewer's positive feedback.
>
> > What is the maximum sequence similarity between the training sequence of ProtBERT and sequences in TrRosetts alignments that were used for testing? Sequences must not overlap have a maximum similarity of let’s say 80%.
>
> We thank the reviewer for this comment. Contact extraction from ProtBERT-BFD is entirely unsupervised. ProtBERT-BFD does not see any contact maps from trRosetta during training. For extracting contacts from ProtBERT-BFD, we do not use the trRosetta alignments, but only the exact reference sequence corresponding to the PDB chain in question.The 500 families used for identifying the top 6 contact-prediction heads from ProtBERT-BFD are disjoint from our 748 test families.
>
> We also note that, typically, the most valuable sequences for improving unsupervised contact prediction are not sequences with high sequence identity to the target, but instead distant homologs with low sequence identity (see “Assessing the utility of coevolution-based residue–residue contact predictions in a sequence- and structure-rich era” Kamisetty et al., 2013). As such, we do not think overlap of close homologs between the trRosetta alignments and BFD pretraining set would introduce biases into our results.
>
> > You describe that you used three sets of families from the TrRosetta dataset (A.4.1). Why did you use only 732 families for testing (set 3)? Were these all families that were not included in the first two sets? How many families do the first two sets include and how similar are families of different sets? Ideally, train, tune, and test families belong to different super families.
>
> We appreciate these questions and have clarified our setup in Section A.6.1 of the revised paper. We use a small set of six families to find settings of learning rate and weight decay for standard attention. We identified a set of ten challenging families for an early version of factored attention and found settings of learning rate and regularization which attain reasonable performance on all families. We have now updated our evaluation procedure to uniformly evaluate across all 748 families.
>
> We note that the risk of overfitting in our setting is extremely low, as we are not training supervised contact prediction.  All single-layer models are trained from scratch for each family (except for the new value-matrix sharing experiment, where values are frozen). Our sweeps were performed to find learning rate and regularization coefficients which did not crash or have serious performance issues. We believe that our models are actually under-optimized compared to the Potts baseline, as the regularization and optimization of those models has been tuned over many years by the structure prediction community.
>
> The reviewer’s suggestion of using superfamilies for model development is an insightful one and we plan to incorporate it into future work.
>
> > You describe in section A.3 how you extracted protein contact maps from the attention maps of ProtBERT. This is an important detail that must be described in the main text. How did you choose the 6 heads? Did you choose them manually or, for example, by training a linear model to predict contacts from attention maps and using the weights for identifying important heads, or computing the weighted average of attention maps?
>
> Thank you for pointing out this omission, and we agree that this merits further explanation. We have added a Section 3.3 titled Extracting Contacts in which we describe our procedure for selecting the heads from ProtBert-BFD. We also provide a table of precisions for ProtBERT heads in Table 2 of Section A.5. We select the six best individual heads whose attention maps had the top average contact precision on 500 families randomly selected from the trRosetta dataset and not in the set of 748 test families. We extract contacts from ProtBert-BFD by averaging the LxL attention maps from these six heads, then symmetrizing additively.

---

> > ### Author Response · Authors · 2020-11-24
> > **Response to Reviewer 2 (Continued)**
> >
> > > Section 3.2, ‘x = E_seq(x_i) + E_pos(i)’: How did you compute positional embeddings and why do and add embeddings instead of concatenating them?
> >
> > > Section 3.2, ‘We treat the positional embedding E_pos as an overall summary per-position information’.  Please describe more clearly what this summary is.
> >
> > In the single-layer attention model, we use a learned positional encoding and sequence embedding. Each single-layer model is trained separately on the aligned positions of a family, so we take the positional encoding to represent the information carried by each position in the MSA. This motivates our contact extraction procedure of using only positional encoding to extract contacts for an MSA. We sum the sequence and positional encodings to make our single-layer attention directly comparable to the Transformer and we believe concatenation would be interesting to explore for follow-up work. We provide these details in Section A.2.
> >
> > > Section 4, first paragraph: The L of the precision at L metric is not the sequence length but the number of top sequences. You describe L as being both.
> >
> > We appreciate this comment from the reviewer. We implement precision at L as defined for the CASP competition, which uses length for L when selecting top predicted contacts. See “Assessment of contact predictions in CASP12: Co-evolution and deep learning coming of age” by Schaarschmidt et al 2018 and “Assessing the accuracy of contact predictions in CASP13” by Shrestha et al 2019. We have clarified our presentation of precision at L in the main text and added these citations to provide context.
> >
> > > Figure 6 is not discussed. Instead of showing this figure, I suggest quantifying the correlation depending on the number of heads by computing and discussing the Spearman correlation.
> >
> > We agree that this result received insufficient discussion in the original manuscript. We now include plots of both pearson and spearman correlation with Potts weights as Figures 11 and 12 in the supplement.  Note that we have also expanded this experiment to compute weight correlations on the full set of 748 families.
> >
> > > Rives et al 2020 ‘Biological structure and function emerge…’ have recently shown in addition to Vig et al that protein contact can be predicted from attention maps, which must be also pointed out in the ‘Background’ section.
> >
> > We thank the reviewer for this suggestion. In our new Background section “Supervised Contact Prediction,” we have now cited Rives et al 2020, mentioning that they used Transformer embeddings as input features to linear projections and deep residual networks for supervised contact prediction

---

### Official Review · AnonReviewer1 · 2020-10-27
**Very interesting work but lack of some clarifications**

**Rating:** 5
**Confidence:** 5

**Review:**

Recently, some researchers tried to apply attention models into the protein field, using self-supervised learning to predict protein contacts. In this work, the author attempt to build the connection between such works and the old-school model, Potts model. By simplifying some operations within the attention model, the author managed to build an analog between the simplified model and the Potts model. The analog is intuitive and easy-to-understand. The authors further compare the simplified model and the Potts model on 748 protein families, showing that they are similar. Or probably the simplified attention model is even better. This is an interesting work. However, I also have a number of concerns. The advantages and disadvantages are listed below.

Pros:
1. The manuscript is concise and easy-to-understand.
2. The idea is intuitive and reasonable, with experimental support.

Cons:
1. The analog between the simplified attention model and the Potts model is intuitive but not rigorous. The authors claim that they provide a theoretical connection between the two models. However, that part is not strong enough, without proof.
2. There are two assumptions in this work, which make the simplified model different from the attention models that the previous researchers used. Firstly, they train the model on multiple sequence alignment instead of the raw sequences. If they train the model on the raw sequences, the performance is unacceptable, as shown in Figure 16, which is consistent with the previous research. Secondly, they removed the sequence embedding in queries and keys. This simplification makes the model only consider the statistical pattern in the MSA. To me, this one is a too strong assumption.
3. The running time and hardware comparison is missing. If the single layer of attention is comparable to the Potts model, not outperform it significantly, while it would take much more time to train, the researchers would need to think twice if they want to use the attention model.
4. The ablation study makes me feel that the results are on the opposite of the conclusion. Here is my logic. With the above two assumptions, the attention model can achieve similar performance as the Potts model, or a little bit better. However, when we train on the unaligned sequences, which is the usual case that we would use the attention model, the performance becomes unacceptable. Then why we want to use the more expensive attention model? The attention model in the NLP field is a different story. Those models are refreshing the STOA performance all the time. However, in the protein field, the attention model can still only achieve comparable performance as the classific models, after a two-year study. They seldom outperform classic algorithms. The results in this manuscript are consistent with the previous research. So I am not convinced regarding the conclusion in the abstract:
"Taken together, these results provide motivation for training Transformers on large protein datasets."
5. The potential audience of this paper would be those who are specialized or interested in bioinformatics and protein.

---

> ### Author Response · Authors · 2020-11-24
> **Response to Reviewer 1**
>
> > The manuscript is concise and easy-to-understand.
> > The idea is intuitive and reasonable, with experimental support.
>
> We appreciate these kind comments.
>
> > The analog between the simplified attention model and the Potts model is intuitive but not rigorous. The authors claim that they provide a theoretical connection between the two models. However, that part is not strong enough, without proof.
>
> We have reorganized Section 3 (“Methods”) to clarify the formulation of Factored Attention as a pairwise MRF. In our new treatment, we directly define factored attention using the energy functional, rather than as a simplification of attention. We hope this clarifies that the simplified attention model is a pairwise MRF by definition. We also believe this new exposition conveys the modeling assumptions in factored attention much better, and we are grateful to the reviewer for this feedback.
>
> > There are two assumptions in this work, which make the simplified model different from the attention models that the previous researchers used. Firstly, they train the model on multiple sequence alignment instead of the raw sequences. If they train the model on the raw sequences, the performance is unacceptable, as shown in Figure 16, which is consistent with the previous research. Secondly, they removed the sequence embedding in queries and keys. This simplification makes the model only consider the statistical pattern in the MSA. To me, this one is a too strong assumption.
>
> We chose to make these assumptions in order to understand how these particular properties of protein families impact performance in isolation and to give a concrete example of models that live between the Transformer and Potts. Our goal is not to propose factored attention as an isolated model which outperforms all others, but to increase understanding of the Transformer’s success and highlight new avenues for model development. We have considerably reworked the methods and discussion to better communicate this goal and make our stance clear. We believe this has significantly improved the quality of our exposition, so we thank the reviewer for this comment.
>
> We agree with the reviewer that our assumptions are considerably stronger than the assumptions made by a pretrained Transformer. We would also note that MSAs are readily available in the unsupervised contact extraction setting where Potts models are currently state-of-the-art.
>
> > The running time and hardware comparison is missing. If the single layer of attention is comparable to the Potts model, not outperform it significantly, while it would take much more time to train, the researchers would need to think twice if they want to use the attention model.
>
> We appreciate the reviewer’s emphasis on computational efficiency. We have added discussion about performance tradeoffs to our section on standard attention. Further, we have provided a detailed study of throughput for all models on various MSA lengths in Section A.3. This includes a supplemental table giving batches/second at various lengths for Potts, factored attention, and single-layer attention. In addition to throughput, we now discuss gains in parameter efficiency more clearly in the results section. These show that factored attention can model the 748 test protein families with 11 billion fewer parameters than Potts (a 91% reduction).

---

> > ### Author Response · Authors · 2020-11-24
> > **Response to Reviewer 1 (Continued)**
> >
> >
> > > The ablation study makes me feel that the results are on the opposite of the conclusion. Here is my logic. With the above two assumptions, the attention model can achieve similar performance as the Potts model, or a little bit better. However, when we train on the unaligned sequences, which is the usual case that we would use the attention model, the performance becomes unacceptable. Then why we want to use the more expensive attention model? The attention model in the NLP field is a different story. Those models are refreshing the STOA performance all the time. However, in the protein field, the attention model can still only achieve comparable performance as the classific models, after a two-year study. They seldom outperform classic algorithms. The results in this manuscript are consistent with the previous research. So I am not convinced regarding the conclusion in the abstract: "Taken together, these results provide motivation for training Transformers on large protein datasets."
> >
> > These comments from the reviewer have been very helpful in helping us improve the paper. We agree wholeheartedly with the reviewer that attention models applied to proteins have not succeeded until they have pushed state-of-the-art in ways not previously imaginable. The purpose of our paper is to provide a clear argument that this is possible, but we do not claim to have achieved it yet.  In our update of the paper, we have clarified that the remarkable aspects of ProtBERT’s performance are not improved precisions, but the fact that ProtBERT was not trained with any protein family labels, data clustering, or even knowledge of the existence of protein families, all of which are available to Potts. We believe that understanding how this is possible through pretraining is an important task for the ML community. Our main contribution is identifying that modeling interactions within families by attention exists on a continuum which includes Potts models. We then use single layer models to identify a set of explicit properties of protein families, such as common amino acid interactions and sparse contacts, and show that attention leverages them for single families. The fact that our single layer models trained on MSAs match the performance of both Potts and ProtBERT validates our claims that there are natural modeling assumptions that benefit attention. These experiments, along with successful sharing of frozen value matrices across hundreds of protein families, provides evidence that development of sophisticated multifamily models is fertile ground for future exploration.
> >
> > > The potential audience of this paper would be those who are specialized or interested in bioinformatics and protein.
> >
> > We believe that our work provides a point of entry into protein modeling for the broader representation learning community, and helps focus future work. This is particularly relevant to those not already specialized in bioinformatics or protein ML, since we suggest that there is novel ML work to be done beyond adapting BERT variants to protein data.
> >
> > Our work also provides a constructive exploration of the capabilities of Transformers applied to proteins. Whereas BERTology-based work tries to probe the weights of a Transformer and disentangle what it’s doing, we take a principled approach of simplifying Transformers into a few key elements and showing that it still succeeds. We think the success of this kind of analysis on protein data shows that protein representation learning can contribute insights to the study of attention.

---

### Official Review · AnonReviewer3 · 2020-10-28

**Rating:** 6
**Confidence:** 4

**Review:**

Summary:
This paper explores the connection between the classic Potts model-based approaches and modern Transformer-based approaches for protein contact map prediction. To this end, the authors introduce a simplified variation of the attention layer called factored attention, and show that a single layer of factored attention performs operations similar to those performed by the Potts model-based methods.

Pros:
- The paper attempts to connect classic and modern approaches to protein contact map prediction, which might be interesting to the people working in this field. The evidence presented (simplifying attention layer so that the equations look similar to the classic methods, numerical results of the simplified attention layer close to the classic methods) is reasonably convincing.
- The topic of the paper is quite timely, there has been a lot of interest recently in modelling proteins using the latest NLP techniques.
- The paper is well written. I appreciate the effort put in by the authors to define basic protein terminologies which might not be obvious to readers without biology background.

Cons:
- The contributions of the paper would have been more interesting if the proposed modifications of the attention layer led to increased prediction performance of models which are representative of the state-of-the-art. Specifically, if retraining ProtBERT-BFD using the modified attention layer led to further improvement in performance, that would have been a solid contribution.
- Are MRF models really that competitive for contact map prediction? From what I understand, deep neural networks have been far better at this task for quite some time now. At multiple places in the paper, the authors give the impression that MRF models are close to state-of-the-art.
- In the last paragraph of the introductory section, the idea of encoding the MSAs is introduced which seemed interesting. However, from what I understood from the rest of the paper, the queries and keys are extracted solely based on the position of the amino acid. Is that right? If so, does the position correspond to the position in the sequence or in the MSA? Are the actual alignments used in any of the results in the paper? Please clarify.

Comments:
- Section 3.1: "each edge" should have a capital e.
- Section 3.3, specifically the part where you show that factored attention is a pairwise MRF, is too brief. Given that this is a main contribution of the paper, it would be worthwhile to explain this connection in a more detailed manner.

---

> ### Author Response · Authors · 2020-11-24
> **Response to Reviewer 3**
>
> > The paper is well written. I appreciate the effort put in by the authors to define basic protein terminologies which might not be obvious to readers without biology background.
>
> We are grateful for this positive feedback from the reviewer.
>
> > The contributions of the paper would have been more interesting if the proposed modifications of the attention layer led to increased prediction performance of models which are representative of the state-of-the-art. Specifically, if retraining ProtBERT-BFD using the modified attention layer led to further improvement in performance, that would have been a solid contribution.
>
> We thank the reviewer for these comments. The goal of our paper, as reflected in our updated draft, is to highlight the potential of models between the two extremes of Potts and Transformers for learning powerful features from protein databases. We don’t present factored attention as an isolated modeling advance, but as a tool for exploring how the modeling assumptions made by attention leverage underlying signal in protein data.
>
> In light of this goal, we have expanded our results to highlight advantages of factored attention that stem from modeling assumptions. Our experiments show that factored attention can do as well as Potts with substantially fewer parameters, in some cases using only a handful of heads. We have also shown that factored attention can train on all 748 families using only one shared set of amino acid features (value matrices), indicating the potential for increased parameter sharing within families.
>
> We would also like to clarify that composing multiple layers of factored attention can not be done simply like with attention, since position and sequence can not be disentangled after the first layer. We believe finding ways to compose modified attention layers presents an interesting avenue for future work and have mentioned this in the Discussion.
>
> > Are MRF models really that competitive for contact map prediction? From what I understand, deep neural networks have been far better at this task for quite some time now. At multiple places in the paper, the authors give the impression that MRF models are close to state-of-the-art.
>
> We appreciate this question from the reviewer and have added a new Background section titled “Supervised Structure Prediction” to help resolve this confusion. Existing deep neural network approaches for contact prediction take a supervised approach, training with pairs of the form (MSA, contact map). All approaches evaluated in this paper are trained only on sequence with no supervised signal from known contact maps. MRF-based features, also known as “coevolutionary features” in the literature, are essential  inputs to the supervised deep neural networks mentioned by the reviewer. (See the reviews of CASP 11 and 12 performance for the importance of coevolutionary features in neural network performance: Monastyrskyy et al., 2016 and Schaarschmidt et al., 2018)
>
> > In the last paragraph of the introductory section, the idea of encoding the MSAs is introduced which seemed interesting. However, from what I understood from the rest of the paper, the queries and keys are extracted solely based on the position of the amino acid. Is that right? If so, does the position correspond to the position in the sequence or in the MSA? Are the actual alignments used in any of the results in the paper? Please clarify.
>
> We have clarified these questions in the updated draft, and we believe it has greatly strengthened the paper. We use position in the aligned sequence for our positional encoding, rather than position in the unaligned sequence. Factored attention computes its queries and keys using only this MSA positional encoding, while the single layer of attention uses both position and sequence. All results for single-layer models (Potts, factored attention, and single-layer attention) are from training on MSAs except for the single ablation study shown in Figure 21.
>
> > Section 3.1: "each edge" should have a capital e.
>
> We appreciate this comment and have fixed it in the manuscript.
>
> > Section 3.3, specifically the part where you show that factored attention is a pairwise MRF, is too brief. Given that this is a main contribution of the paper, it would be worthwhile to explain this connection in a more detailed manner.
>
> This comment was very helpful to us and we have followed the reviewer’s advice in rewriting our Methods section. We have moved the mathematical discussion to the supplement and now focus on the underlying modeling assumptions that give rise to factored attention. We hope this greatly clarifies how factored attention differs from Potts as a pairwise MRF and also explains why factored attention (and attention more broadly) is a natural model class for protein families.

---

### Official Review · AnonReviewer4 · 2020-11-04
**Too basic and lacks compelling use case**

**Rating:** 5
**Confidence:** 4

**Review:**

This manuscript describes a connection between Potts models and attention as implemented in modern transformers. The authors then present an attention model in which positional encodings are defined as one-hot vectors indicating fixed positions in the multiple sequence alignment and train single layer attention models. These models, unsurprisingly, perform similarly to Potts models without APC correction for contact prediction. The methods section is somewhat confusingly written. I think the factored attention model would benefit from being described on it’s own terms rather than in connection with typical multiheaded attention, especially because the isolation of position encodings and amino acids at those positions dramatically simplifies the understanding of W_Q, W_K,  and W_V. The authors also spend a long time describing well known methods, but without providing additional insight. The connection between the Potts model and attention described in this paper should be obvious to those who already understand attention models and Potts models and the empirical results of the factored attention model don’t make this approach seem compelling. In the discussion, the authors make several broad future speculations. Some of these would be interesting contributions and I encourage the authors to develop this work further. Maybe factored attention could be promising for better capturing dependencies between positions for deeper transformers on MSAs, but it isn’t likely that this work will be of broad interest to the machine learning community. This manuscript seems better suited to a workshop or other specialized venue. Some specific comments on this work follow below.
1.	In the factored attention model, the authors use one-hot encoding of the position index as the position encoding. This is equivalent to learned position embeddings as in BERT which is worth mentioning.
2.	The authors discuss single-site potentials as a difference between Potts models and single layer attention models and then show a comparison of attention models with and without single-site potentials showing little difference. However, attention models already implicitly have single-site potentials which arise from the positional encoding input features. Granted, this is not the case for the factored attention model where single-site potentials seem to have more effect, though in the negative direction.
3.	The authors state that “The ability of factored attention to capture similar contacts to Potts without use of APC suggest that it may be more suitable for protein design.” I don’t follow this conclusion. If the factored attention model performs equivalently to the Potts model alone and worse than the Potts model with APC correction, why would it be more suitable for protein design?
4.	 What makes the single-layer attention or factored attention models compelling for protein modeling? What problems do these models solve that are not better solved by the Potts model or traditional transformers?

What would raise my score:
1.	Present a compelling use case for the factored attention model. What questions can be answered (or better answered) with this model over the Potts model or other alternatives? One idea is to use the factored attention model as the layers in a full deep transformer model and see if this architecture can improve tasks where MSA training data is available.

Edit: I have increased my score in light of the response and manuscript edits. The manuscript is improved, but I think the method still needs more development. There are a number of interesting pieces but the final picture of an improved protein model is not fully resolved.

---

> ### Author Response · Authors · 2020-11-24
> **Response to Reviewer 4**
>
> > The methods section is somewhat confusingly written. I think the factored attention model would benefit from being described on it’s own terms rather than in connection with typical multiheaded attention, especially because the isolation of position encodings and amino acids at those positions dramatically simplifies the understanding of W_Q, W_K, and W_V.
>
> We found these comments from the reviewer very helpful when rewriting our Methods section for the updated draft. We have moved the discussion linking factored attention and attention to Section A.2, and have instead presented both factored and single-layer attention by adding assumptions to Potts models step-by-step. We agree with the reviewer that a direct presentation of factored attention is much clearer. One of our goals in this paper is to highlight why attention makes use of natural properties of protein family data, and we believe this new exposition contributes to that. We hope the reviewer finds the new structure much improved.
>
> > The connection between the Potts model and attention described in this paper should be obvious to those who already understand attention models and Potts models and the empirical results of the factored attention model don’t make this approach seem compelling.
>
> We agree that the connection between Potts and attention described in this paper is mathematically simple, but we see this as a clear advantage of our paper. Our goal in introducing factored attention is to break down various aspects of the Transformer in a protein-specific setting, relate them to existing state-of-the-art Potts models, and understand how much these particular assumptions impact performance. We also believe that, while those of us who have spent the effort to understand both attention models and Potts models do find the connection in this paper rather self-evident, it is of broad use to the community for it to be spelled out and validated empirically in an application domain of significant scientific importance. We have also expanded our results section to more clearly lay out insights brought by factored attention which are not readily apparent from either Potts models or ProtBERT. We believe this context further demonstrates the value of our approach.
>
> > In the discussion, the authors make several broad future speculations. Some of these would be interesting contributions and I encourage the authors to develop this work further.
>
> We appreciate this comment from the reviewer. As part of our rewrite, we have considerably shortened our discussion and focused on the aspects of multifamily models and pretrained Transformers not explored by our analysis. Speculations about the role of APC and the quality of sequences sampled from factored attention have been removed to help increase focus.
>
> > Maybe factored attention could be promising for better capturing dependencies between positions for deeper transformers on MSAs, but it isn’t likely that this work will be of broad interest to the machine learning community. This manuscript seems better suited to a workshop or other specialized venue.
>
> We appreciate this comment and have worked to better clarify what aspects of this work are of interest to the broader ML community. We believe protein representation learning is a topic of considerable interest in ML and that our work suggests new avenues, beyond pretraining Transformers, to the community.
>
> > In the factored attention model, the authors use one-hot encoding of the position index as the position encoding. This is equivalent to learned position embeddings as in BERT which is worth mentioning.
>
> This is a helpful observation from the reviewer which we have added to the discussion in Section A.2.
>
> > The authors discuss single-site potentials as a difference between Potts models and single layer attention models and then show a comparison of attention models with and without single-site potentials showing little difference. However, attention models already implicitly have single-site potentials which arise from the positional encoding input features. Granted, this is not the case for the factored attention model where single-site potentials seem to have more effect, though in the negative direction.
>
> We are grateful for this observation from the reviewer that the positional encoding input to single-layer attention model creates a single-site term implicitly. We have added discussion on this point to Section A.2, “Implicit single-site term in single-layer attention.”

---

> > ### Author Response · Authors · 2020-11-24
> > **Response to Reviewer 4 (continued)**
> >
> > > The authors state that “The ability of factored attention to capture similar contacts to Potts without use of APC suggest that it may be more suitable for protein design.” I don’t follow this conclusion. If the factored attention model performs equivalently to the Potts model alone and worse than the Potts model with APC correction, why would it be more suitable for protein design?
> >
> > We have removed this sentence as part of focusing our discussion.
> >
> > This particular discussion was based on the paper “An evolution-based model for designing chorismate mutase enzymes” by Russ et al (2020), which includes a discussion on how sequences sampled from Potts models do not match underlying MSA statistics, indicating poor sample quality. Sampling does not involve APC, so models that can improve on performance of Potts without APC could be fruitful for sequence generation.
> >
> > > What makes the single-layer attention or factored attention models compelling for protein modeling? What problems do these models solve that are not better solved by the Potts model or traditional transformers?
> >
> > We are grateful to the reviewer for these important questions. The goal of our paper, as reflected in our updated draft, is not to present factored attention as an isolated advance over existing Potts models or Transformers, but instead to demonstrate that there exists a huge unexplored space between these two ends of the spectrum. We are, to our knowledge, the first work to clearly explain that Potts models and Transformers represent two extremes for modeling databases of protein families, and we see our single-layer models as an essential part of demonstrating that there exist interesting models in the middle.
> >
> > As part of addressing the reviewer’s concerns, we have reworked our experiments to more clearly highlight interesting phenomena of factored attention not immediately available to either Potts or Transformers. Our new results more carefully show that factored attention is able to use relatively few heads for recovering L or L/5 contacts and that factored attention is able to successfully match the performance of Potts models on all test families using one frozen set of amino acid features. These results demonstrate that the parameter sharing introduced by factored attention can drastically reduce the number of parameters needed compared to Potts, while making more explicit assumptions about protein families than Transformers. We have also laid out empirical advantages of ProtBERT-BFD over our single layer models as evidence that even more powerful multifamily models based on scientifically grounded assumptions may exist.
> >
> > > Present a compelling use case for the factored attention model. What questions can be answered (or better answered) with this model over the Potts model or other alternatives? One idea is to use the factored attention model as the layers in a full deep transformer model and see if this architecture can improve tasks where MSA training data is available.
> >
> > We reproduce our main contributions stated in the global comment:
> > 1. We show that attention can be linked to Potts models using purely biological assumptions about protein data, and provide evidence that these assumptions are borne out in protein structural data.
> > 2. Empirical evaluations of factored attention’s performance show that these assumptions lead to competitive performance with Potts models, and show that they lead to increased parameter-efficiency on long families. We also present the ability to tie value matrices across all families as evidence that multifamily hierarchical structure is readily accessible even to single attention layers.
> > 3. We show that ProtBERT-BFD can learn contacts competitively with Potts models over a wide range of protein families. This builds on recent work from Vig et al, but the work of Vig et al does not carefully compare contact prediction metrics with an optimized Potts model implementation. We believe this is an encouraging result that suggests pretraining Transformers on proteins merits further work.
> > 4. We contrast the performance and parameter efficiency of ProtBERT-BFD and factored attention to suggest the existence of even richer unrecognized hierarchical structure exploited by pretrained Transformers and not leveraged by either Potts, factored attention, or single-layer attention.

---

### Author Response · Authors · 2020-11-24
**Global Comment to All Reviewers**

We are very grateful to the reviewers for their comments and suggestions, which have helped us significantly to improve our paper in many aspects. We have worked hard to incorporate feedback from all reviewers in our updated draft. We have provided detailed responses to each reviewer, but would like to outline a few overall changes to the paper.

__Methods Section:__ All reviewers commented that the methods section could be more accessible. We have now reworked it to more clearly state the modeling assumptions of both factored and single-layer attention. This also clarifies the connection to Potts models. We hope the reviewers find this section much improved.

__Improving Loss Functions:__ We realized the regularization we used for factored attention did not match the regularization used for Potts. We have fixed this and updated all results. We provide a discussion of regularization in the Methods section and Section A.4 of the Appendix.

__Expanded Exploration of Hyperparameters:__ We have expanded our experiments on the impact of number of heads and head size, running each configuration on the entire set of 748 families rather than 10. The results section has been updated and expanded accordingly. We have also highlighted a specific example of interest in Figure 5 which shows that only 4 heads can be used to extract contacts for a particular family.

__Added Experiment on Shared Amino Acid Features:__ We have added an experiment on parameter sharing across families. Our paper mostly focuses on the capacity of attention to share parameters across positions within a single family, but we believe an initial exploration of sharing across hundreds of families highlights the exciting work yet to be done on attention-based hierarchical models of many protein families.

__Clarifying Contributions and Position:__ Many reviewers asked us to clarify the use-case and goals of introducing factored attention. Our paper addresses the broader questions around pretraining large models on databases containing thousands or more protein families. Unlike in NLP, there remains considerable debate about if these models are succeeding and if it is worth continuing to develop them further, a question also raised by Reviewer 1. Our paper contributes to this broader question in protein representation learning in four major ways:

1. We show that attention can be linked to Potts models using purely biological assumptions about protein data, and provide evidence that these assumptions are borne out in protein structural data.
2. Empirical evaluations of factored attention’s performance show that these assumptions lead to competitive performance with Potts models, and show that they lead to increased parameter-efficiency on long families. We also present the ability to tie value matrices across all families as evidence that multifamily hierarchical structure is readily accessible even to single attention layers.
3. We show that ProtBERT-BFD can learn contacts competitively with Potts models over a wide range of protein families. This builds on recent work from Vig et al, but the work of Vig et al does not carefully compare contact prediction metrics with an optimized Potts model implementation. We believe this is an encouraging result that suggests pretraining Transformers on proteins merits further work.
4. We contrast the performance and parameter efficiency of ProtBERT-BFD and factored attention to suggest the existence of even richer unrecognized hierarchical structure exploited by pretrained Transformers and not leveraged by either Potts, factored attention, or single-layer attention.

The goal of our paper is to reframe the discussion around pretraining attention-based models on protein sequences. Critiques of pretraining focus on whether pretraining at scale is effective compared to existing state-of-the-art techniques. We believe our contributions indicate that there is ample room for combining both approaches by building novel hierarchical models of protein family databases. We also take the position that such hierarchical models will heavily involve attention, possibly modified attention mechanisms designed specifically for protein data.

---

### Decision · Program_Chairs · 2021-01-07
**Final Decision**

**Decision:**

Reject

**Comment:**

The paper shows a connection between Potts model and Transformers and uses the connection to propose a factored attention energy to use in an MRF. Results are shown, using this energy based on factored attention. Also, pretrained BERT models are used to predict contact maps as a comparison.
The reviewers found the paper interesting from a protein structures prediction point of view, but from a machine learning perspective their opinion was that the paper does not offer a coherent, compelling method that is very novel, and the connection between Potts and an energy based attention model is not that overwhelming.  In addition the presentation was somewhat circuitous.

The authors made improvements to the paper over the course of the review, which is appreciated, but the method presented does not match the target for an ICLR paper in terms of methodological contributions.